# Interventional Effects of Edible Bird’s Nest and Free Sialic Acids on LPS-Induced Brain Inflammation in Mice

**DOI:** 10.3390/nu17030531

**Published:** 2025-01-31

**Authors:** Nan Qian, Chen-Xi Zhang, Guan-Dong Fang, Shuang Qiu, Yu Song, Man Yuan, Dong-Liang Wang, Xiang-Rong Cheng

**Affiliations:** 1School of Food Science and Technology, Jiangnan University, Wuxi 214122, China; qiannan202412@163.com (N.Q.); chenxi.zhang@chinyou.com (C.-X.Z.); fgd12232023@163.com (G.-D.F.); ssongyuu@163.com (Y.S.); 2State Key Laboratory of Food Science and Resources, Jiangnan University, Wuxi 214122, China; 3Hebei Edible Bird’s Nest Fresh Stew Technology Innovation Center, Langfang 065700, China; shuang.qiu@xxdun.com (S.Q.); man.yuan@xxdun.com (M.Y.); 4Central Nervous System Drug Key Laboratory of Sichuan Province, Luzhou 646000, China

**Keywords:** sialic acid, edible bird’s nest, lipopolysaccharides, neuroinflammation

## Abstract

**Objectives:** Our study investigated the effects and mechanisms of edible bird’s nest (EBN) and free sialic acids (SA) on LPS-induced brain inflammation in mice. **Methods**: The experiment divided the mice into four groups: control group (CON), lipopolysaccharide group (LPS), EBN intervention group (EBN, 200 mg/kg/d in dry EBN), and sialic acid intervention group (SA, dosage was calibrated based on the concentration of sialic acid in EBN). **Results**: The results showed that LPS caused a decrease followed by upregulation in body weight in female mice, and EBN exhibited renal protective effects. In the Morris water maze, the learning and memory abilities of mice in the LPS group first declined and then recovered. At the same time, the escape latency improved in the EBN and SA groups. In the Open field test, both the EBN and SA groups exhibited anti-anxiety and anti-depressive effects. Immunohistochemistry in the hippocampus showed significant cell damage in the LPS group, while the damage was alleviated in the EBN and SA groups. LPS promoted the expression of TICAM1 and MYD88 in the NF-κB pathway, while both the EBN and SA groups could inhibit the expression of TICAM1. **Conclusions**: The study has found that both EBN and SA exhibited noteworthy anti-inflammatory effects, indicating that the main active component in EBN that provides neuroprotective effects is SA. The bound SA in EBN confers additional effects, supporting the development of prevention and treatment strategies for brain inflammation.

## 1. Introduction

Brain inflammation, as a key link in the pathogenesis of many neurological diseases, has become one of the research hotspots in neuroscience today. Among the many factors that induce brain inflammation, lipopolysaccharide (LPS), as a major component of the cell wall of Gram-negative bacteria, can effectively mimic the state of bacterial infection [1]. LPS can cause dysbiosis of the mouse gut microbiota, disrupt the gut barrier, and trigger gut inflammation [2]. And then activate the body’s immune response, triggering a complex and intense inflammatory response in the brain [3]. This inflammatory response involves over-activation of microglia, massive release of pro-inflammatory cytokines, alteration of blood-brain barrier permeability, and neuronal dysfunction [4,5], which often causes severe damage to the central nervous system, significantly affecting the normal physiological functions of the body, such as cognition and behaviour, and posing a heavy burden on the well-being of patients and a serious challenge to public health.

Sialic acid (SA), a class of acidic aminoglycans widely present in living organisms, has recently attracted much attention in medical and biological research [6]. There are many members in the SA family, commonly including N-acetylneuraminic acid (Neu5Ac) and its derivatives [7]. The basic structure of SA is a nine-carbon glycoside derivative formed by the condensation reaction of a pyruvic acid molecule with a mannosamine molecule. From a morphological point of view, SA exists in different structural forms, and the SA derived from edible bird’s nest (EBN) primarily exists in a bound state [8], which is usually connected with sugar chains in biological macromolecules such as glycoproteins and glycolipids through specific glycosidic bonds. For example, in EBN, SA is often combined with proteins to form glycoprotein complexes, which show complex glycan chain branching in their structure, and the SA group is at the end of the glycan chain [9]. This structural feature endows it with unique biological properties, enabling it to recognize and bind organisms, and also influences the metabolization, transition, and interaction with other biomolecules [10]. EBN positively affects intestinal health [11,12] and regulates the body’s brain and liver through metabolites produced by intestinal flora [13,14,15]. EBN is beneficial to women’s health and can prevent and treat neurological disorders in menopausal women [16].

On the other hand, free SA is not covalently bound to other macromolecules and exists as a monomer in the organism. Compared to glycoprotein-bound SA, free SA molecular structure is relatively simple and independent. It has an intact basic structure of acidic aminosugars and free carboxyl and amino functional groups that exhibit unique chemical activities in solutions. It is more likely to interact with small molecules or ions. Its diffusion and transport mechanism through the cell may vary from that of glycoprotein-bound sialic acid.

SA has been shown to play important roles in various biological processes, including immune regulation, cell signalling and neural development, and some studies have suggested that they have potential anti-inflammatory properties [17]. EBN is associated with the improvement of cognitive functions. Studies indicate that it has neuroprotective effects on Alzheimer’s disease, with the ability to inhibit neuroinflammation [18] and neuronal cell death [19]. Therefore, EBN benefits cognitive functions, which may be attributed to its antioxidant and anti-inflammatory activities [20]. Studies have shown that SA is the main neuroprotective active ingredient in EBN, and both free and bound SAs have been found to be present in EBN in comparable amounts. However, systematic and in-depth studies have yet to be conducted on different forms of SA, especially glycoprotein-bound SA and free SA from EBN, in addressing LPS-induced brain inflammation in mice and the specific mechanisms that affect brain inflammation development.

Given this, the present study was designed to construct an LPS-induced brain inflammation model in mice. By using free SA and EBN in mice, this work attempted to compare if free or bound SA confer greater bioactivity in EBN, namely from neurobehavioural, inflammation-related indexes, and inflammation pathways standpoint. It is expected to reveal further the mechanism of different forms of SA on brain inflammation and to provide a valuable theoretical and experimental basis for developing sialic acid-based preventive and therapeutic strategies for brain inflammation-related diseases.

## 2. Materials and Methods

### 2.1. Materials and Reagents

Edible bird’s horn (from Malaysia, provided by Beijing Yungshutang Biotechnology Co., Ltd., Beijing, China); sialic acid (Neu5Ac, prepared by immobilised enzyme method, Wuhan Kostan Biotechnology Co., Ltd., Wuhan, China); lipopolysaccharides (LPS, L2280, Sigma, Saint Louis, MI, USA); PBS buffer tablets (Solebo, Beijing, China); physiological saline (Shuanghe Pharmaceuticals Co. Ltd., Anhui, China); GFAP primary antibody (A14673, ABclonal, Wuhan, China); IBA-1 primary antibody (Ab178846, Abcam, Shanghai, China); secondary FITC goat anti-rabbit (111-095-003, Jackson, Wuhan, China); anhydrous ethanol, NaHSO_4_, o-phenylenediamine hydrochloride, glucose, phenol, concentrated sulfuric acid, methanol, etc. were purchased from Sinopharm Chemical Reagent Co. Ltd., Shanghai, China; tumor necrosis factor alpha (TNF-α), interleukin-1β (IL-1β), interleukin-6 (IL-6) commercial kits were purchased from Xiamen Huijia Co. Ltd., Xiamen, China; acetonitrile for liquid chromatography was all LC grade and purchased from Sinopharm Chemical Reagent Co. Ltd., Shanghai, China.

### 2.2. Experimental Animals

A total of 32 7-week-old female C57BL/6J mice (SPF grade) were purchased from Jiangsu Jicui Pharmachem Biotechnology Co., Nanjing, China. The experiment was approved by the Experimental Animal Welfare and Ethics Committee of Jiangnan University, with the approval number JN. No. 20230415c0800705.

### 2.3. Instruments and Equipment

The following equipment was used: a constant temperature oscillation incubator (TXA-120, Taicang Qiangle Experimental Equipment Co., Ltd., Taicang, China); a liquid chromatograph (LC-20AT, SHIMADZU, Kyoto, Japan); EF-C18 Bio column (Galaksil, Wuxi, China); a 96-well plate enzyme labeller (EPOCH1, Bio Tek Instruments Inc, Winooski, VT, USA); a panoramic section scanner (PANNORAMIC DESK/MIDI/250/1000, 3DHISTECH, Budapest, Hungary); an optical microscope (NIKON ECLIPSE E100, Nikon, Tokyo, Japan); and a decolourisation shaker (SK-0180-E, SCILOGEX, LLC, Los Angeles, CA, USA).

### 2.4. Determination of Sialic Acid Content

Referring to Wang Hui [21], the O-phenylenediamine (OPD) pre-column derivatisation method was used for the determination with appropriate modifications. Free SA was determined in a water bath at 50 °C protected from light for 2.5 h, and the temperature was changed to 4 °C to continue derivatisation for 48 h. HPLC working conditions: EF-C18-Bio column, SPD detector, column temperature and detector cell temperature of 30 °C, mobile phase acetonitrile: water = 5: 95, flow rate: 1.0 mL/min, detection wavelength: 230 nm, injection volume: 20 μL, isocratic elution: 45 min.

### 2.5. Animal Grouping and Biological Sample Collection

Mice were housed in the barrier facility of Jiangnan University Laboratory Animal Centre (SYXK-(Su)-2021-0056) and were subjected to a cycle of light and darkness every 12 h. Temperature: 20~26 °C, humidity: 40~70%, under an artificial light/dark cycle for 12 h. All mice were randomly divided into four groups, which were set as blank control group (CON), lipopolysaccharide group (LPS), EBN intervention group (EBN), and sialic acid intervention group (SA), with eight mice in each group. The CON and LPS groups were given 0.2 mL of 0.9% saline by gavage daily from week 2. The intervention groups were given equal volumes of EBN homogenate (200 mg/kg/d in dry EBN) and SA solution (dosage was calibrated based on the concentration of sialic acid in EBN) [18,19,22] for 7 weeks, respectively. From the last week onwards, 0.2 mL of 0.9% saline was injected intraperitoneally daily in the CON group, and an equal volume of LPS solution (500 μg/kg) was injected intraperitoneally in the other groups for 7 day (d) [23]. As shown in Figure 1, behavioural tests were carried out six hours after daily intraperitoneal injection for 7 d. After the end of all behavioural tests, the mice were fasted overnight and anesthetized with isoflurane inhalation after weighing. They were euthanized using the cervical dislocation method, and then immediately dissected. Organ samples were collected on ice and weighed, and plasma, serum, and organ tissues were frozen at −80 °C.

### 2.6. Behavioural Experiment

#### 2.6.1. Morris Water Maze (MWM) 

The Morris water maze apparatus consisted of a circular white water tank (120 cm in diameter and 50 cm in height) placed in the center, surrounded by an opaque cloth. A stationary platform (10 cm in diameter) was submerged 1 cm below the water surface, which was dyed white to obscure the platform. The water temperature was maintained at 25 °C.

On the 1st–5th day of the MWM, mice underwent spatial localization training. Each day, mice were introduced into the maze from four different quadrants facing the pool wall and were given 60 seconds (s) to spontaneously search for the hidden platform. The latency and path taken by the mice to reach the platform were recorded. If a mouse failed to find the platform within 60 s, it was guided to the platform and allowed to stay there for 15 s. In such cases, the escape latency was recorded as 60 s. To prevent the mice from locating the platform through inertia, the entry points were varied each day.

On the 6th day, a spatial exploration test was conducted to assess memory consolidation. The platform was removed from the maze, and the diagonal of the platform’s original position was used as the new entry point (180° from the original platform location). Mice were allowed to swim freely for 60 s. Their trajectories were monitored, and the number of times each mouse crossed the former platform position, as well as the time spent in the target quadrant, were recorded [24].

#### 2.6.2. Open Field Test (OFT)

The open field test was conducted in a white, rectangular box with dimensions of 50 cm × 50 cm × 40 cm (length × width × height). A camera was mounted on the top of the box to record the trajectories of the mice. The test was performed in a quiet, gray environment to minimize external disturbances. Mice were gently placed on the edge of the box facing the wall and allowed to move freely for a period of 5 min. Their trajectories were recorded throughout this time. After each mouse’s behavioral data were collected, the box was sprayed with 75% alcohol to eliminate any residual odors, ensuring that the behavior of subsequent mice would not be influenced by the scent of previous animals. The mice’s behavior was analyzed based on several parameters, including central area dwell time, total distance, freezing time, and standing number [25].

### 2.7. Measurement of Inflammatory Factor Levels

Levels of cytokines TNF-α, IL-1β, and IL-6 in mouse brain (hippocampus, cortex) homogenates were determined using commercially available Elisa kits (Xiamen Huijia Co. Ltd., Xiamen, China), following the instruction guide. Enzyme-labeled reagents were added to a 96-well plate according to the instructions, followed by incubation and washing steps. After the color reaction, 1.0 M H_2_SO_4_ solution was added to stop the reaction. Immediately thereafter, the absorbance values of each well were measured at a wavelength of 450 nm, and the cytokine concentration was calculated based on the obtained standard curve equation.

### 2.8. Nissl Staining and Immunofluorescence Staining of Hippocampus

Fresh mouse brains were preserved in paraformaldehyde fixative, paraffin-embedded, and made into tissue sections, which were later deparaffinized to water and stained by adding toluidine blue for 2–5 min. After washing in water, 0.1% glacial acetic acid was added to differentiate the reaction, and the reaction was terminated by washing with tap water, and dried in the oven. Xylene was added for 10 min and the sections were sealed with neutral gum. The slices were examined microscopically under a light microscope and images were collected after Nissl staining.

After dewaxing paraffin-embedded brain sections to water, they were subjected to antigen retrieval in a microwave oven. Following cooling, the sections were decolorized with PBS (pH 7.4). The sections were then air-dried and blocked with 3% BSA for 30 min at room temperature. Next, primary antibodies against GFAP and IBA-1 (diluted 200-fold and 1000-fold, respectively, in PBS) were added dropwise to the sections, followed by overnight incubation at 4 °C in the dark. After decolorization and drying, secondary antibodies specific to the primary antibodies (diluted 400-fold in PBS) were applied and incubated at room temperature in the dark. Subsequently, DAPI was added dropwise and incubated for 10 min at room temperature. Finally, the sections were mounted with a fluorescence quenching sealer, and panoramic images of the hippocampus were captured using a tissue section scanner. The mean optical density (MOD) values of green fluorescence in the images were analysed using Image-Pro Plus 6.0 software. Calibration standards are converted to optical density values. Measurements selection “Area” and “IOD”. Colors selection “Histogram Based”, and threshold setting “0–250”. The mean optical density was calculated according to Formula (1), as follows:(1)$$\mathrm{MOD}=\frac{IOD}{\mathrm{Area}}$$

### 2.9. Quantitative Real-Time PCR (qPCR) Analysis

RNA was extracted from hippocampus tissue using a column extraction method according to the instructions. Total RNA was then reverse-transcribed into cDNA using the HiScript^®^ III RT Super Mix reverse transcription kit (Vazyme Biotechnology Co., Ltd., Nanjing, China). Quantitative real-time reverse transcription polymerase chain reaction (RT-qPCR) amplification was performed using the 2× Taq Pro Universal SYBR qPCR Master Mix (Vazyme Biotechnology Co., Ltd., Nanjing, China). The β-actin was used as an endogenous control for normalizing tar-get gene mRNA expression. The relative expression levels of target genes were calculated using the 2^−△△ct^ method. The primer sequences (designed by NCBI GenBank) are listed in Table 1.

### 2.10. Data Processing and Analysis

The data results conforming to normal distribution were expressed as mean ± standard deviation of the mean (Mean ± SEM). GraphPad Prim 8 software was used to carry out statistical analysis of the data and to draw pictures. Data conforming to normal distribution were analysed using non-parametric tests, one-way ANOVA with post hoc two-by-two comparisons using Tukey’s model for data in 3 or more groups; a *t*-test was used for data in 2 groups. The KW rank sum test was chosen for parametric tests, and a threshold of *p* < 0.05 was set to distinguish whether the groups were statistically different from each other.

## 3. Results

### 3.1. Characterisation of Different Forms of Salivary Acids

The treatment was performed using the OPD pre-column derivatisation method and quantified by HPLC. The free SA content of edible bird’s horn was 7.35 ± 0.16%, and the total SA content was 14.07 ± 0.03%, consistent with the literature reports [26].

### 3.2. Analysis of Body Weight Organ Index

The changes in body weight of female mice in each group during the modelling time are shown in Figure 2A. Compared with the control group, it was clear that the body weight of female mice under continuous LPS invasion continued to decrease on days 2–3. It was significantly lower than that of the CON group, with a gradual increase in body weight from day 4 onwards. There was no significant difference between the LPS and the CON groups on day 5, which is consistent with the transient nature of LPS-induced mold formation reported by previous research [3,27], and suggests that the body can promote the restoration of homeostasis, but the recovery of body weight does not indicate fully repaired metabolic disorders. Notably, the EBN and SA groups showed similar changes in body weight as the LPS group, with no significant difference, indicating that they did not have any significant restorative ability on the body weight of female mice invaded by LPS.

The protective effects of structurally different SAs of EBN on the organs are shown in Figure 2B–D. The liver, kidney, and spleen indexes became significantly more prominent in the LPS group compared with the control group (*p* < 0.001). These indexes were reduced to different degrees in the EBN and SA groups compared with the LPS group. Among them, the EBN group showed a significant kidney index difference compared to the LPS group (*p* < 0.05), and the kidney index recovered by 5.05%. It can be shown that both EBN and free SA have partial protective effects on the organs, especially EBN, which exhibited more obvious protective effects on the kidneys.

### 3.3. Behavioural Analysis

#### 3.3.1. MWM Analysis

Excessive inflammatory responses in the brain can adversely affect learning and memory abilities [28]. Therefore, the MWM is used to examine mice’s learning and memory capabilities through the cognitive assessment of mammalian rodents. As shown in Figure 3A, the Morris water maze experiment can reveal a significant difference in latency between the 1–4 d LPS group and the CON group (*p* < 0.05), indicating a significant decrease in spatial learning and memory ability. As shown in Figure 3B–D, the Escape period of EBN and SA groups on day 2–3 was significantly lower than that of LPS group (*p* < 0.05), and the Escape period on day 2 was decreased by 25.85% and 21.51%, suggesting their ameliorative effect. On day 4, SA group increased to the same level as LPS group, but the latency of EBN group was still significantly lower than LPS group (*p* < 0.05).

Figure 3E–F reveals no significant difference between the groups on the test metrics in the MWM in the probe experiment on day 6, indicating little difference in spatial memory ability between the groups. For the LPS group, this indicated that their spatial memory capacity was restored from day 5. Seven consecutive days of intraperitoneal injection of model-ling in the LPS group resulted in a significant decrease in the learning memory capacity of the female mice on days 1–4, while the capacity was restored from day 5 onwards. This was similar to the trend of body weight. 

#### 3.3.2. OFT Analysis

The OFT can reflect the degree of tension, anxiety, or depression of female mice in an unfamiliar environment. Excessive inflammatory responses in the brain are associated with depression [29]. The proportion of time spent in the central area by female mice in each group during the total time is shown in Figure 4A, which measures of anxiety-like behaviour. There was no significant difference between the LPS group and the CON group, whereas SA group showed a significant increase in the proportion of time spent in the central area, suggesting that SA has an anxiolytic effect. The total distance moved in each group within the open field experiment is shown in Figure 4B, and the LPS group showed a significant decrease in the distance compared to the control group (*p* < 0.05), suggesting that LPS induced depression-like behaviour in female mice. In the SA group, the distance moved by the female mice was significantly increased (*p* < 0.05) compared to the LPS group, indicating that SA can better exert antidepressant effects.

As shown in Figure 4C, the resting time of female mice in each group reflects the anxiety level and locomotor ability of female mice. Compared with the CON group, LPS significantly increased the resting time (*p* < 0.01), suggesting that LPS made female mice more anxious, depressed, and less capable of exercise. Compared with LPS group, the rest time of female mice in EBN group and SA group was reduced by 50.42% and 48.23%, respectively. It indicated that both structurally different SAs in EBN alleviated the depression-like behaviour. The number of erections of female mice in each group is shown in Figure 4D, reflecting the exploratory behaviour of the mice and the degree of curiosity about the new environment. It was found that there was no significant difference between the groups in the number of erections compared with the CON group. Taken together, at day 7, LPS-induced female mice still showed some degree of depression-like behaviour. Moreover, both EBN and SA could alleviate the depression-like behaviour of female mice.

### 3.4. Analysis of Brain Inflammatory Factors

Intraperitoneal injection of LPS promotes transient and strong expression of inflammatory cytokines in various brain regions of mice, such as IL-1β, TNF-α, and IL-6 [30]. As shown in Figure 5, the LPS group had different effects on the expression of inflammatory factors: IL-1β is a pro-inflammatory factor, and the LPS group increased the expression of IL-1β compared with the control group, which can be seen that LPS induced inflammation in female rats. The expression of IL-1β in the EBN and SA groups was reduced to different degrees compared with that of the LPS group. The expression of TNF-α is a pro-inflammatory factor. Compared with CON group, LPS had no significant effect on the expression of TNF-α in mice brain.

Meanwhile, no significant difference was observed between the EBN and SA groups compared with the LPS group. IL-6 is a pro-inflammatory and anti-inflammatory factor, and IL-1β and TNF-α inflammatory factors mainly activate its expression. Compared with the CON group, the LPS group reduced the expression of IL-6 inflammatory factors in the brain, especially in the cortical area (*p* < 0.01). The EBN and SA groups were alleviated compared with the LPS group, and the expression of IL-6 inflammatory factor was recovered by 15.95% and 75.57%.

### 3.5. Immunohistochemistry of the Hippocampus

The results of Nissl staining revealed the morphological changes of neurons in hippocampal CA1, CA2, and DG regions. CA1 plays a key role in memory consolidation and is considered an important link in converting short-term memory into long-term memory. Damage to the CA1 region can seriously impact spatial memory [31]. Damage to the CA2 region reduces social memory ability, affects subsequent memories’ storage and retrieval, and is involved in a certain degree of emotional memory [32]. The DG region helps the brain distinguish similar but different inputs to avoid memory confusion. This region is closely related to the plasticity of the brain and its ability to learn and adapt to new environments and things [33].

As shown in Figure 6, in the CA1, CA2, and DG regions, the single-cell structure in the CON group had a complete outline, clear nuclei, and a high number of intact nerve cells. In contrast, the number of intact neurons was significantly reduced in the LPS group, with cell lysis in the marginal layer, diffuse cellular debris, and thick nucleoli. There may be cellular morphology alterations in the LPS group due to inflammatory response [34]. Unlike the LPS group, neurons increased in the EBN and SA groups, with some apoptotic cells but reduced fragmentation. The morphology and arrangement of cells in both groups were more ordered and tightly packed than in the LPS-treated group, with less cellular debris in the hollow space, and the cellular structure appeared more intact. The EBN and SA groups may have some cytoprotective effects, which could alleviate the cellular damage induced by the LPS group to a certain extent. In response to LPS, EBN, and SA, the CA1, CA2, and DG regions did not show noticeable specific differences, which may indicate that these treatments’ effects on different hippocampus regions are somewhat consistent. This experiment found that SAs in EBN alleviated the release of inflammatory factors induced by LPS and reduced neuronal structural damage in the hippocampus by combining the levels of inflammatory factors in the brain.

Ionized calcium binding adapter molecule 1 (IBA-1) is a microglia-specific intracellular protein and is therefore commonly used as a marker to determine microglia specificity. Glial fibrillary acidic protein (GFAP) can be used as a biomarker for astrocyte proliferation during CNS injury. As illustrated in Figure 7, the immunofluorescence staining and quantitative analysis of fluorescence intensity revealed that hippocampal IBA-1 staining was enhanced in LPS-treated mice, indicating increased microglial activation. Additionally, the number of GFAP-positive cells, indicative of astrocyte activation, was also elevated in these mice. Both EBN and SA interventions attenuated the increase in IBA-1-positive and GFAP-positive cells within the hippocampus. Among them, the intervention with EBN demonstrated a better efficacy compared to SA in reducing the activation of these glial cells.

### 3.6. NF-κB Inflammatory Signalling Pathway

NF-κB is a nuclear transcription factor with pleiotropic regulatory effects widely present in various cells. In inflammatory responses, the NF-κB signaling pathway plays a key regulatory role. Researchers have found that inhibiting the activation of key NF-κB signaling targets can reduce the expression of inflammatory factors and alleviate inflammation [35]. Under normal conditions, NF-κB is bound to its inhibitory protein IκB in the cytoplasm, remaining inactive [36,37,38,39]. As can be seen in Figure 8A, the expression of the TICAM1 gene was significantly elevated under LPS treatment conditions (*p* < 0.05). It was suggested that LPS could trigger inflammatory responses by activating the Toll-like receptor 4 (TLR4) signalling pathway. The TICAM1 gene may play an important role in this inflammatory response, and its high expression may be related to the immune defense mechanism. Compared with the LPS group, the TIC AMI gene expression was lower under treatment in the EBN and SA groups (*p* < 0.05), which implied that EBN and SA inhibited the expression of the TICAM1 gene and might have an anti-inflammatory effect.

In Figure 8B, compared with the control group, NF-κB p65 gene expression in LPS group was significantly increased (*p* < 0.05). NF-κB is an important transcription factor in inflammatory responses, immune responses, and cell survival. LPS stimulation resulted in the degradation of IκκB by phosphorylation and NF-κB release, which enabled the NF-κB p65 subunit to enter the nucleus and initiate the transcription of a series of genes related to inflammatory response, immune response, and cell survival. The relatively low expression of NF-κB p65 gene in EBN and SA treatments compared to LPS may indicate that they inhibit the NF-κB signaling pathway activation, and thus have potential anti-inflammatory properties.

Figure 8C showed significantly increased expression of the MYD88 gene under LPS treatment (*p* < 0.05). MYD88 is a key junction protein in the TLR4 signalling pathway in LPS-induced inflammatory response. It can activate downstream NF-κB and other inflammation-related signalling pathways. MYD88 gene expression was significantly decreased in EBN (*p* < 0.01) and SA (*p* < 0.05) groups, which indicated that EBN and SA can reduce the LPS-induced inflammatory response by inhibiting the expression of MYD88.

The expression levels of Iκκβ gene in each group were shown in Figure 8D. Normally, NF-κB binds to the IκB protein in an inactivated state in the cytoplasm. When the cell is stimulated, the Iκκβ is activated and phosphorylates the IκB protein, resulting in the degradation of the IκB protein, NF-κB release, and allows it to enter the nucleus to initiate transcription. Compared with CON group, the expression of Iκκβ gene was increased in LPS group, but there was no significant change, which may indicate that LPS activate the NF-κB pathway with little effect on Iκκβ gene transcription level. This may be because regulation of Iκκβ occur more at the post-translational level, such as its own phosphorylation and activation, rather than at gene transcription.

In Figure 8E, LPS treatment led to the elevation of the expression of the COX-2 gene, an inducible enzyme mainly in the nucleus of cells. COX-2 is an inducible enzyme that mainly impacts inflammation and tumour development. LPS can induce the expression of COX-2 by activating NF-κB and other signalling pathways to produce inflammatory mediators such as prostaglandins. The expression of the COX-2 gene was reduced to a certain extent under EBN and SA treatment, which indicated that EBN and SA has the anti-inflammatory effect of inhibiting COX-2 expression.

CHUK gene expression was shown in Figure 8F. CHUK is part of the Iκκ complex and is involved in the activation of NF-κB. There was no significant change in CHUK gene expression under each treatment, which may indicate that these treatments had little effect on CHUK gene transcription.

## 4. Discussion

LPS caused a significant increase in organ indices, while EBN and SA groups reduced these indices. This may be due to LPS disrupting the body’s metabolism and physiological functions. SA is the main active component in EBN that suggests neuroprotection in mice [14]. Studies suggest that various plant polysaccharides (mycelia selenium polysaccharides, Hericium Erinaceus polysaccharide, etc.) have a positive impact on inflammation-related diseases in mice [40,41,42], indicating that polysaccharides could serve as a novel anti-inflammatory agent. In contrast, EBN can alleviate the damage to organs caused by inflammation, reducing the organ stress response and pathological changes caused by inflammation, although not affecting weight recovery. This may be related to its high SA content which can regulate immune responses, inhibit inflammatory mediator release, and reduce oxidative stress damage that created a relatively stable internal environment for the organs. Among them, EBN is particularly prominent in protecting the kidneys. The bound SA from EBN may confer additional effects on it, which is consistent with previous studies. Researchers have found that bound SA is beneficial for regulating the intestinal probiotics of pregnant women [8], and protein-bound Neu5Ac exhibits higher prebiotic activity than its free monomeric form [43]. 

MWM reveal that the LPS group experienced a significant decline in learning and memory in the early stages. Although there is a tendency for recovery in the later stages, it is due mainly to limited compensatory mechanisms of the body. This experiment revealed that supplementation with different forms of SA could alleviate the decline in learning and memory ability induced by LPS, playing a role in cognitive protection. The EBN and SA groups significantly improved anti-inflammatory, anti-oxidative, and neuroprotective properties, inhibiting inflammatory factors, enhancing neurotrophic factors, repairing neurotransmitter metabolism, and regulating the endocrine system. On day 4 of MWM, EBN still had good spatial cognitive abilities. However, the SA group had declined to the same level as the LPS group, which further confirms that bound SA in EBN exhibits higher anti-inflammatory activity than SA groups (free SA). The impact of different forms of SA on learning and memory functions has been studied to some extent, and it is believed that SA is closely related to the function and stability of the nervous system. SA is a major component of gangliosides and glycoproteins in the brain, such as neural adhesion molecules, which play a key role in forming neural synapses and the process of neural central conduction [44]. Memory is closely related to stable synaptic connections and neural conduction. The experimental results from infant rats [45] and piglet [46] models suggest that SA is likely a conditional nutrient during the rapid growth phase of the brain. The chemical structure of polysaccharides is the basis of their biological activity. The proteins that bind to polysaccharides are different, making it difficult to characterize them to obtain an accurate structure [47]. Neural adhesion molecule (NCAM) is a glycoprotein on the surface of cells, which can be modulated by the addition of SA chains, namely salivation. Sialylated NCAM (PSA-NCAM) can affect the nature of cell adhesion, which in turn affects neuronal growth, migration, and synapse formation. The SA in EBN may theoretically be able to exert a salivating effect on NCAM, but this effect depends on some factors, including bioavailability, dose, and metabolic pathway. Overall, the SA in EBN may have potential salivating effects on neural adhesion molecules.

OFT reveal that LPS group exhibited depression-like behavior. However, there is no clear evidence that SA directly causes significant changes in GABA, serotonin or dopamine levels. SA’s anxiolytic effects may be achieved through mechanisms such as modulation of the function of neurotransmitter systems, enhancement of GABA neuron activity, or activation of AMP-activated protein kinase (AMPK). Other research also found that the total distance traveled, average speed, and frequency of rearing behavior were significantly reduced in the LPS group of mice (*p* < 0.0001). At the same time, the intervention with dihydrolipoic acid improved these indicators (*p* < 0.01) and significantly alleviated the depressive state [48]. A new compound, QTC-4-MeOBnE, significantly increased the grooming frequency in LPS mice during the sucrose splash test (*p* < 0.001) and reduced the immobility time in the forced swim test (*p* < 0.001) [25]. It is speculated that the improvement in depressive behavior is related to restoring BBB permeability and reducing inflammation mediated by NF-κB in the hippocampus and cortex. Given the immunomodulatory and anti-inflammatory activities of structurally different SAs, it is speculated that EBN and its potential active component, namely SA may alleviate depression in LPS mice through anti-inflammatory pathways. However, whether the additional effects of bound SA in EBN are caused by the change in the form of SA or by the synergistic effect of the proteins [49,50] bound to it is currently unclear. We require further investigation to specific mechanism of action of SA at the neurochemical level in future studies, which can guide interventions for cognitive disorders, provide targets for mood disorder research, and lead to the development of functional foods or adjunctive medications, promoting the prevention and treatment of brain diseases. 

LPS has varying effects on different inflammatory factors, and the EBN and SA groups can generally reduce the expression of inflammatory factors. The results showed that individual inflammatory factor indexes in LPS female mice did not differ significantly from the CON group. At the same time, EBN and SA had different levels of interventional effects on alleviating inflammation in mice, which indicates that SAs in EBN can alleviate the increase of inflammatory factors in the brain induced by LPS, exerting regulatory activity on neuroinflammation [51]. Research has found that EBN affects changes in gut microbiota, often accompanied by changes in immune cells and inflammatory factors [52], such as the increase of Bacteroides and Akkermansiaceae and the decrease of Firmicutes in the intestinal flora caused an increase in the levels of TNF-alpha and improved immunity [53,54]. In addition, EBN and SA may act on blocking the activation of surface receptors on immune cells (microglia, astrocytes) [39]. Studies indicate that the modeling method of intraperitoneal injection of 500 μg/kg or 700 μg/kg LPS for seven consecutive days results in significant loss of hippocampal neurons in mice [23]. Researchers have pointed out that natural small-molecule substances in food and medicine have neuroprotective effects, and prevention and treatment strategies may improve neurodegenerative diseases [55]. In experiments, SA has shown good anti-inflammatory activity, providing a specific reference value for treating neurological diseases. The EBN and SA groups exert protective effects through anti-inflammatory and anti-oxidative actions, regulating intracellular signaling pathways and promoting cell repair and regeneration. However, studies on the combination of EBN with other neuroprotective drugs are still in the preliminary stage. The combined application of EBN with antioxidant drugs, anti-inflammatory drugs and brain-derived neurotrophins is a future research direction.

LPS activates genes related to the pathways, while EBN and SA inhibit the expression of some genes, showing an anti-inflammatory trend. In summary, LPS showed significant induction of expression on TICAM1, MYD88 and NF-κB p65 genes, which is consistent with previous studies on the mechanism of LPS-induced inflammatory response [56]. TICAM1, an adaptor molecule with Toll-interleukin-1 receptor domains, is activated after Toll-like receptors (TLRs) recognize pathogen-associated molecular patterns, which in turn activates transcription factors such as NF-κB, initiating the expression of genes related to inflammatory responses [57,58]. LPS triggers the inflammatory response by activating the TLR4- MYD88-NF-κB signalling pathway and up-regulating inflammation-related genes. EBN and SA exhibited inhibitory potential in several inflammatory genes. This may imply that EBN and SA have anti-inflammatory effects, which may be related to inhibition of LPS-activated inflammatory signaling pathways, such as TLR4-MYD88-NF-κB. The NF-κB pathway is at the core of inflammation, and LPS activates the TICAM1-dependent pathway through TLR4. By screening key targets in the pathway, small molecule inhibitors can be designed, and functional peptides that mimic the effects of EBN and bound SA can be developed, combined with gene editing technology to reveal the function and regulatory network of pathway genes, leading to the development of precise anti-inflammatory therapies and nutritional health products.

## 5. Conclusions

LPS significantly elevated the organ index, disrupted the metabolic and physiological functions of the organism, and led to the decline in learning and memory abilities and the emergence of depression-like behaviours. However, the organ index was significantly reduced under EBN and SA treatments, which attenuated the damage of inflammation to the organs and ameliorated the LPS-induced decline in learning and memory ability. The EBN group performed better than the SA group in spatial cognitive ability, implying that the bound SA in EBN may have higher anti-inflammatory activity. In addition, LPS significantly induced the expression of inflammatory genes (TICAM1, MYD88, and NF-κB p65), which activated the TLR4-MYD88-NF-κB signalling pathway and triggered an inflammatory response. EBN and SA exerted their anti-inflammatory effects by inhibiting the inflammatory signalling pathway. Although EBN and SA showed significant potential in anti-inflammation and neuroprotection, their specific mechanisms of action require further study. Future studies should focus on the mechanism of action of SA at the neurochemical level, as well as the differences in biological activity between bound and free SAs. In addition, studies on the combined application of EBN with other neuroprotective drugs will provide an important reference for the development of novel anti-inflammatory therapies and nutraceuticals.

## Figures and Tables

**Figure 1 nutrients-17-00531-f001:**
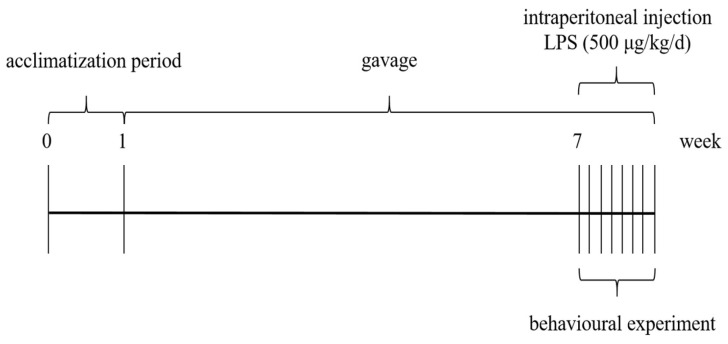
Timeline of animal experiments.

**Figure 2 nutrients-17-00531-f002:**
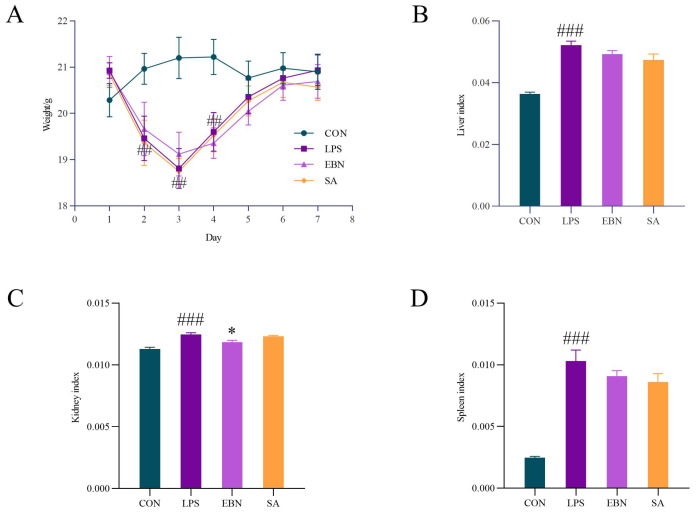
Comparison of body weight changes and organ indices of female mice in each group. (**A**) Body weight changes of female mice in each group during 1–7 days of modelling time. (**B**) Liver index of female mice in each group. (**C**) Kidney index of female mice in each group. (**D**) Spleen index of female mice in each group. Data are expressed as mean ± SEM. ## *p* < 0.01 and ### *p* < 0.001 vs. control group; * *p* < 0.05 vs. LPS group.

**Figure 3 nutrients-17-00531-f003:**
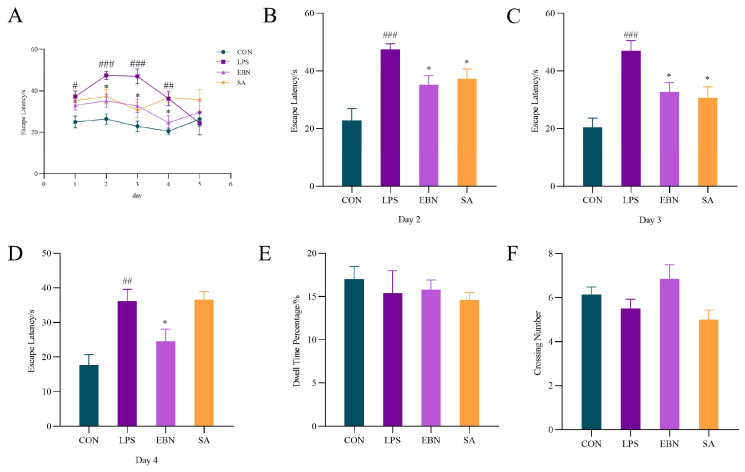
Comparison of behavioural indices in the Morris water maze of female mice in each group. (**A**) Changes in escape latency in female mice of each group during days 1–5 of training. (**B**) Escape latency of female mice in each group on day 2 of training. (**C**) Escape latency of female mice in each group on day 3 of training. (**D**) Escape latency of female mice in each group on day 4 of training. (**E**) Ratio of time spent in the target quadrant by each group of female mice. (**F**) Frequency of female mice passing the target quadrant in each group. Data are expressed as mean ± SEM. # *p* < 0.05, ## *p* < 0.01 and ### *p* < 0.001 vs. control group; * *p* < 0.05 vs. LPS group.

**Figure 4 nutrients-17-00531-f004:**
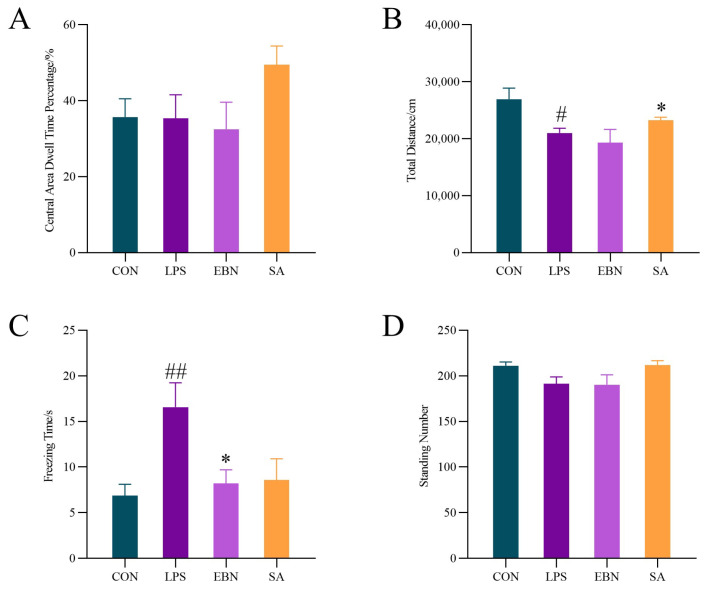
Comparison of behavioral indices of Open field test in female mice in each group. (**A**) Percentage of time spent in the central area by female mice in each group. (**B**) Distance moved within the experiment by each group of female mice. (**C**) Resting time of female mice in each group in the experiment. (**D**) Number of times female mice in each group stood upright in the experiment. Data are expressed as mean ± SEM. # *p* < 0.05, ## *p* < 0.01 vs. control group; * *p* < 0.05 vs. LPS group.

**Figure 5 nutrients-17-00531-f005:**
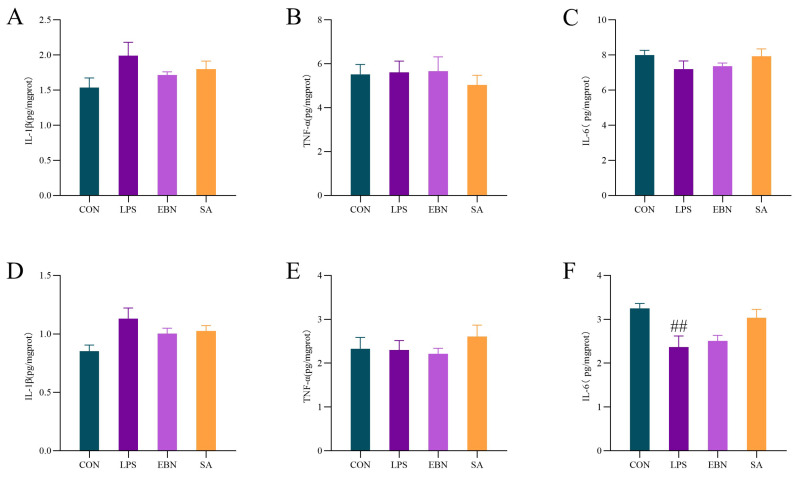
Comparison of inflammatory factors in the brain of female mice in each group. (**A**–**C**) Inflammatory factors (IL-1β, TNF-α, IL-6) in the hippocampus of the brain of female mice in each group. (**D**–**F**) Inflammatory factors (IL-1β, TNF-α, IL-6) content in the cortex of the brain of female mice in each group. Data are expressed as mean ± SEM. ## *p* < 0.01 compared to control.

**Figure 6 nutrients-17-00531-f006:**
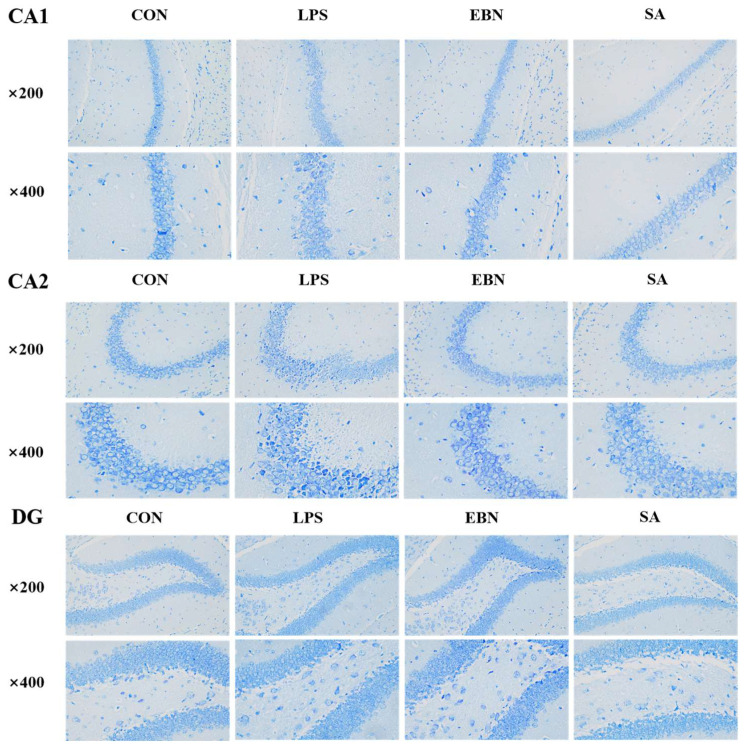
Morphological structure of neurons in CA1, CA2 and DG regions of the hippocampus of female mice in each group (×200, ×400).

**Figure 7 nutrients-17-00531-f007:**
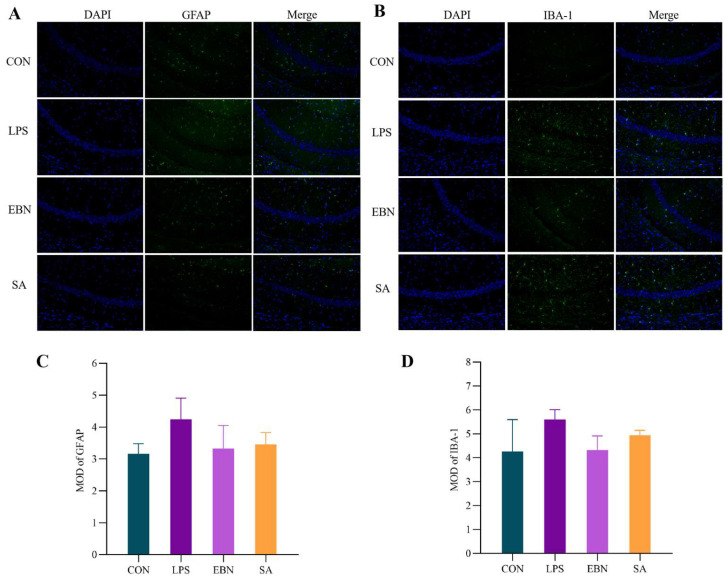
Immunofluorescence staining images and quantitative analysis in the CA1 region of the hippocampus of female mice in each group. (**A**,**B**) Immunofluorescence staining of the CA1 region of the hippocampus with specific antibody labelled GFAP and IBA-1 proteins (×200). (**C**,**D**) Mean optical density in the CA1 region of the hippocampus for specific antibody-tagged IBA-1 and GFAP proteins. Data are expressed as mean ± SEM.

**Figure 8 nutrients-17-00531-f008:**
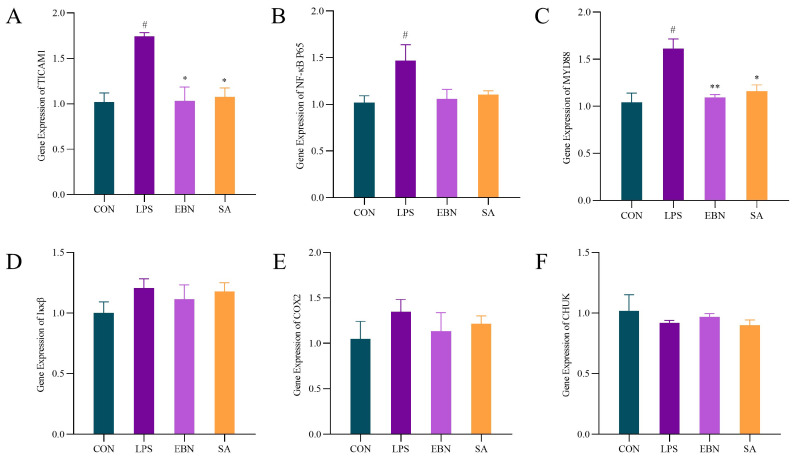
Expression of inflammation-related genes on the NF-κB pathway in each group. (**A**) TICAM1 gene expression in each group. (**B**) NF-κB p65 gene expression in each group. (**C**) MYD88 gene expression in each group. (**D**) IκκB gene expression in each group. (**E**) COX-2 gene expression in each group. (**F**) CHUK gene expression in each group. Data are expressed as mean ± SEM. # *p* < 0.05 vs. control group; * *p* < 0.05, ** *p* < 0.01 vs. LPS group.

**Table 1 nutrients-17-00531-t001:** Gene primer sequence for qPCR.

Gene	Forward Primer (5’-3’)	Reverse Primer (5’-3’)
β-actin	GGCTGTATTCCCCTCCATCG	CCAGTTGGTAACAATGCCAT
TICAM1	CAAGCTATGTAACACACCGCT	TGGTAACCCTAAGGAGACACTG
NF-κB p65	ACTGCCGGGATGGCTACTAT	TCTGGATTCGCTGGCTAATGG
MYD88	CAGGAGATGATCCGGCAACT	CATGCGGCGACACCTTTTC
Iκκβ	GGCACCCAATGATTTGCCAC	TCTAAGAGCCGATGCGATGT
COX-2	TTCCAATCCATGTCAAAACCGT	AGTCCGGGTACAGTCACACTT
CHUK	AAGGCCATTCACTATTCTGAGGT	GTCGTCCATAGGGGCTCTT

## Data Availability

The original contributions presented in the study are included in the article, further inquiries can be directed to the corresponding author.

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
