# Peer review of "Interventional Effects of Edible Bird’s Nest and Free Sialic Acids on LPS-Induced Brain Inflammation in Mice"

_nutrients, 2025, doi:10.3390/nu17030531_

Round 1

Reviewer 1 Report

Comments and Suggestions for Authors

Journal: Nutrients (ISSN 2072-6643)

Manuscript ID: nutrients-3424647

Title: Interventional effects of edible bird's nest and free sialic acids on LPS-induced brain inflammation in mice

The study's findings offer insight on the anti-inflammatory, neuroprotective, and antioxidant properties of edible bird's nest (EBN) and sialic acid (SA) in the treatment of LPS-induced inflammation and cognitive deficits. The study demonstrated that EBN and SA can preserve organs, regulate inflammation (particularly the NF-κB pathway), and improve cognitive and behavioral results. The proven efficacy of bound SA in EBN over free SA opens up a new route for developing functional foods and treatment solutions. Overall, this research serves expand the understanding of natural bioactive chemicals' role in neuroinflammation and brain health.

Abstract:

1.      Line 14's "de-crease" needs to be reworded for readability because it seems to contain an extra hyphen.

2.      Some sentences repeat findings without providing new information, such as the statement of both EBN and SA's neuroprotective effects.

3.      The amounts of EBN and SA given to the mice are not provided in the abstract.

4. The manner of administering LPS (such as dose, frequency, and route) is not included.

5. The specific markers or procedures utilized in immunohistochemistry to determine hippocampus injury are not specified.

Introduction:

6. Line 71–74: "However, systematic and in-depth studies have yet to be conducted on different forms of SA, especially glycoprotein-bound SA and free SA from EBN, in addressing LPS-induced brain inflammation in mice and the specific mechanisms that affect brain inflammation development."

This statement is excessively broad and fails to explain why this comparison is scientific or therapeutically meaningful.

7. Lines 75-80, the explanation appears just vague:

"Given this, the present study was designed to construct an LPS-induced brain inflammation model in mice. This work compared the interventional effects of two different forms of SA in EBN and conducted an in-depth investigation from neurobehavioral, inflammation-related indexes, and inflammation pathways standpoint."

It is lacking in detail about the study's novelty or practical relevance.

2. Materials and Methods

8. Line 132–143: External factors might affect animal behavior, compromising the reliability and reproducibility of behavioral studies. The Morris water maze (MWM) and Open field test (OFT) testing environments are not detailed in depth (for example, room light, noise levels, or time of day).

9. Line 117–129: What is the positive control (e.g., a known anti-inflammatory agent) to validate the experimental system?

Line 149–155: Mention the methodology for quantifying staining results (e.g., software parameters, threshold settings for optical density analysis).

83–93: Ensures reagent consistency and experiment reproducibility.

10. Line 117-129: validates the experimental setup and provides an idea of reference for interpretation.

11. Line 149-155: improves the reliability and repeatability of histology results.

12. Line 248-249: What neurochemical mechanisms are responsible for SA's anxiolytic properties? Are there evident changes in GABA, serotonin, or dopamine levels?

13. Line 233-234: How do EBN and SA affect synapse stability and neural adhesion molecules at the molecular level?

14. Line 303-309: Why are there no significant differences in TNF-α expression between LPS and control groups, despite documented inflammatory responses?

15. Could mixing EBN with other recognized neuroprotective drugs improve its therapeutic efficacy?

Reviewer 2 Report

Comments and Suggestions for Authors

The manuscript brings new information about the health-beneficial effects of edible bird's nest (EDN) and its important component sialic acid (SA) on LPS-induced inflammation in the mouse brain. The structure of the study was chosen logically with regard to the main goal, and the work presents the results of in vivo experiments examining the effects on the cognitive functions (anti-anxiety and anti-depressive effects of EBN and SA) in mice as well as the effects on inflammatory markers in the brain. The results clearly indicated the preventive effect of EBN and SA when is part of food on inflammatory diseases in the brain of LPS-induced mice, which can also induce chronic inflammatory diseases in humans. Authors confirmed that bound SA in EBN exhibits higher anti-inflammatory activity than free SA.

The abstract and introduction are clearly written and the English language is grammatically correct. However, other parts of MS, especially Methods and Results, need to be significantly revised. One of the main rules when writing a scientific article is to describe the used methodologies in sufficient detail so that they can be verified/repeated by other authors. However, the authors described some procedures insufficiently resp. incorrectly. I will provide suggestions how to improve this part:

Line 94: 2.2 Experimental animals: Provide more information in accordance with ARRIVE guidelines.

Line 114: 2.5 Animal grouping and biological sample collection

119 „(LPS), EBN intervention group (EBN), and salivary acid intervention group (SA), with…” Abbreviation SA was used for sialic acid and in this paragraph is used for salivary acid. Correct this. The same is on line 123 “the concentration of salivary acid in EBN) [18, 19, 22] for 7 weeks, respectively.” Explain abbreviation “d” as it is used for first time in this paragraph.

Line 121: „EBN homogenate (200 mg/kg/d in dry EBN).  How was dry EBN administered to mice? Add that SA was given to mice per os.  

Line 127: What does it mean? “the mice were executed”… Add more information how mice were killed, whether anesthesia was administered and what preparation was used to induce it and whether it caused death or another form of killing was applied.

Line 144: 2.7 Measurement of inflammatory factor levels

 “Levels of cytokines TNF-α, IL-1β, and IL-6 in mouse brain (hippocampus, cortex) homogenates were determined using commercially available Elisa kits“. Add information how samples of tissue were prepared, in what kind of solutions and how the concentration of cytokines was calculated. I found in Figures that it was calculated for mg of host proteins. Add information whether soluble proteins were detected and what method was used. Add information of name of KITs and producer. After stopping reaction with acid, OD is usually measured at 450 nm and not 490 nm. Can you explain it in more detail?

Line 149: 2.8 Nissl staining and immunofluorescence staining of hippocampus.

Line 150: “Fresh brains of mice were sectioned by paraffin and stained with Nissl's dye. Brains were sectioned in paraffin, titrated with DAPI stain”. This sentence is completely unclear. From the results, it is likely that the tissue was processed in the form of paraffin blocks after fixation. It is necessary to rewrite the entire procedure and clearly declare what was used as a fixation agent and how the tissue was processed Processing in an alcohol series and subsequent embedding in paraffin blocks?). What was used for subsequent staining? Commercial kit or classic protocol (need to cite procedure). Why DAPI fluorescent titration was used? Images shown in Fig. 6 seemed to be taken in light microscope. Explain this and add necessary information for readers.

Line 157: 2.9 Quantitative Real-time PCR (qPCR) analysis

This paragraph is described inadequately and very superficially and grammatically incorrect because the individual steps are given in the indefinite article. It is necessary to completely rewrite this part according to the standard rules. It is necessary to state how the RNA was isolated from the tissues, whether the concentration was measured and how much was used for cDNA preparation. Whether KIT or individual components were used. It is necessary to describe the material and manufacturer (Kit?) used for qPCR (containing SYBRE green) and how the expression was calculated. Authors cite (26) Zhou, F., et al., Identification of novel NF-κB transcriptional targets in TNFα-treated HeLa and HepG2 cells. Cell Biology International, 535 2017. 41(5): p. 555-569” , but this is not methodological paper. Have authors used 2-DDCt method? What type of gene was used for normalization? It is not mentioned in the text. It is also necessary to indicate whether the primers were designed by the authors and, if so, by which program. If not, then indicate the citations where the sequence was mentioned for the first time.

There is missing paragraph about organ characteristics and what the “indices” shown in Results mean? How were they calculated?

The RESULTS section is described in sufficient detail and the results are clearly summarised in the graphs. However, this part is written rather non-standardly, because the authors have attached texts in each part of the results that need to be moved to the Discussion part. Here are some examples:

Lines 224-234:  move sentences to Discussion

Line 253: correct “de-pression-like” to depression-like

274-284:  move sentences to part Discussion

Lines 296 – sentences: “Research has found that consuming bird's nest affects 296 changes in gut microbiota, often accompanied by changes in immune cells and inflammatory factors[36], such as the increase of Bacteroides and Akkermansiaceae and the decrease of Firmicutes in the intestinal flora caused an increase in the levels of TNF-alpha and improved immunity[37, 38].“ move sentences to part Discussion.

Line 306-310: “The results showed that individual inflammatory factor indexes in LPS female mice.....“ move sentences to part Discussion.

 Lines 316: 3.5 Immunohistochemistry of the hippocampus

If possible enlarge images and insert arrows /arrowheads to point the pathological changes after LPS: „the number of intact neurons was significantly reduced in the LPS group, with cell lysis in the marginal layer, diffuse cellular debris, irregular contraction of the residual cells, and thick nucleoli, apoptotic cells....“ How authors can see on histological sections using this simple stain irregular contractions of cells. Why no more specific immunohistological staining using specific antibodies was not applied?

Lines 342-347:” Studies indicate that the modeling method of intraperitoneal injection of 500 μg/kg or 700 μg/kg LPS for seven consecutive days ..........“ -Move these sentences to Discussion.

Fig.7: Expression of inflammation-related genes on the NF-κB pathway in each group. „ It quite surprising that „gene expression“ for all 6 genes was only up to 2. Therefore, It is necessary to describe in part Methods details how these tests were performed and calculated.

Lines 403-414: “In summary, LPS showed significant induction of expression on TICAM1, MYD88 and NF-κB p65 genes, which is consistent with previous studies on the mechanism of LPS-...“ Move this sentences to Discussion.

By default, the statistical average in the results and legend is given as mean ± SD or SEM and not vice versa as is written in this MS “Data are expressed as SEM ± mean”. Overall, the graphs are of good quality, but the y-axis legends need to be enlarged and adjusted.

DISCUSSION MUST be completely revised and texts from part Results with references incorporated.

Comments on the Quality of English Language

English language must be improved. 
